# Modelling Aboveground Biomass Carbon Stock of the Bohai Rim Coastal Wetlands by Integrating Remote Sensing, Terrain, and Climate Data

Shaobo Sun [1,2,3], Yafei Wang [4], Zhaoliang Song [1,2,3,*], Chu Chen [5], Yonggen Zhang [1,2,3], Xi Chen [1,2,3], Wei Chen [1,2,3], Wenping Yuan [6], Xiuchen Wu [7], Xiangbin Ran [8], Yidong Wang [9], Qiang Li [1] and Lele Wu [1]

1 Institute of Surface-Earth System Science, School of Earth System Science, Tianjin University, Tianjin 300072, China; shaobo.sun@tju.edu.cn (S.S.); ygzhang@tju.edu.cn (Y.Z.); xi_chen@tju.edu.cn (X.C.); chenwei19@tju.edu.cn (W.C.); liqiang3513@tju.edu.cn (Q.L.); lelewu1997@tju.edu.cn (L.W.)

2 Tianjin Key Laboratory of Earth Critical Zone Science and Sustainable Development in Bohai Rim, Tianjin University, Tianjin 300072, China

3 Critical Zone Observatory of Bohai Coastal Region, Tianjin University, Tianjin 300072, China

4 Key Laboratory of Regional Sustainable Development Modeling, Institute of Geographic Sciences and Natural Resources Research, Chinese Academy of Sciences, Beijing 100101, China; wangyafei@igsnrr.ac.cn

5 Tianjin Institute of Surveying and Mapping Co., Ltd., Tianjin 300381, China; chenml.18s@igsnrr.ac.cn

6 Guangdong Province Key Laboratory for Climate Change and Natural Disaster Studies, Zhuhai Key Laboratory of Dynamics Urban Climate and Ecology, School of Atmospheric Sciences, Sun Yat-sen University, Zhuhai 510245, China; yuanwp3@mail.sysu.edu.cn

7 State Key Laboratory of Earth Surface Processes and Resource Ecology, Beijing Normal University, Beijing 100875, China; xiuchen.wu@bnu.edu.cn

8 Research Center for Marine Ecology, First Institute of Oceanography, Ministry Natural Resources, Qingdao 266061, China; rxb@fio.org.cn

9 Tianjin Key Laboratory of Water Resources and Environment, Tianjin Normal University, Tianjin 300387, China; wangyidong@tjnu.edu.cn

* Correspondence: zhaoliang.song@tju.edu.cn; Tel.: +86-022-2740-5053

**Abstract:** Remotely sensed vegetation indices (VIs) have been widely used to estimate the aboveground biomass (AGB) carbon stock of coastal wetlands by establishing Vis-related linear models. However, these models always have high uncertainties due to the large spatial variation and fragmentation of coastal wetlands. In this paper, an efficient coastal wetland AGB model for the Bohami Rim coastal wetlands was presented based on multiple data sets. The model was developed statistically with 7 independent variables from 23 metrics derived from remote sensing, topography, and climate data. Compared to previous models, it had better performance, with a root mean square error and *r* value of 188.32 g m$^{-2}$ and 0.74, respectively. Using the model, we firstly generated a regional coastal wetland AGB map with a 10 m spatial resolution. Based on the AGB map, the AGB carbon stock of the Bohai Rim coastal wetland was 2.11 Tg C in 2019. The study demonstrated that integrating emerging high spatial resolution multi-remote sensing data and several auxiliary metrics can effectively improve VIs-based coastal wetland AGB models. Such models with emerging freely available data sets will allow for the rapid monitoring and better understanding of the special role that "blue carbon" plays in global carbon cycle.

**Keywords:** aboveground biomass; ANPP; coastal wetlands; remote sensing; Sentinel satellite; carbon stock

## 1. Introduction

Coastal wetlands, including salt marshes, tidal marshes, estuaries, mangroves, and seagrasses, demonstrate higher atmospheric $CO_2$ sequestration rates and contribute significantly to the mitigation of climate warming when compared to most other terrestrial ecosystems despite their small global coverage [1]. Carbon storage in these coastal wetlands is referred to as "coastal wetland blue carbon" [2], which is accumulated in biomass on a

short-term scale and in sediments in the long term [3]. However, a rapidly changing climate (specially warming and sea-level rise) and strong human activities (e.g., reclamation and eutrophication) have weakened the blue carbon sink and will further damage it in the future [4].

Vegetation biomass is becoming severely reduced in most global coastal areas. About one-third of global mangrove, seagrass, and salt marsh areas have been lost over the past several decades [3]. The natural coastal wetlands were reduced by more than 53% in China during the period 1957–2017 [5,6]. These reductions in coastal wetlands largely decrease the blue carbon sink through reducing aboveground biomass (AGB). The decreased AGB reduces the burial rates of organic matter [7,8] and increases the net ecosystem exchange (i.e., decreasing carbon sink) [9]. Thus, to evaluate carbon sequestration and storage in coastal wetlands and to mitigate the reduction in blue carbon sink, it is essential to quantify the AGB of coastal wetlands at a large spatial scale [10].

Satellite remote sensing provides a practicable and flexible tool to estimate vegetation AGB at a large scale. It has been used extensively to model the AGB of grasslands [11], forests [12], and wetlands [10,13] since the 1980s by relating the field AGB measurements to remote sensing vegetation indices (VIs) such as the normalized difference vegetation index (NDVI) [14] and the enhanced vegetation index (EVI) [15]. However, compared to the successful use of AGB modeling for forests and grasslands, it is much more difficult to accurately estimate AGB with VIs in coastal wetlands, as coastal wetlands mainly consist of estuary wetlands, tidal marshes, salt marshes, and mariculture ponds; are fragmented in terms of spatial distribution; and have high spatial heterogeneity [13]. The AGB of coastal wetlands is also strongly affected by seasonal variations as well as annual hydrology patterns and human activities.

Many researchers have attempted to estimate the AGB of coastal wetlands using models related to satellite remote sensing VIs and field AGB measurements [10,16–18]. These models can be divided into two types, machine learning (ML)-based models and VIs-based linear regression models. Compared to VIs-based models, ML-based models often perform better when predicting the AGB of coastal wetlands [10,17]. However, ML-based models always need a larger number of field samples to train and validate the models and are much more complex to apply than VIs-based models. Acquiring sufficient AGB samples from coastal wetlands is an enormous and costly project. In contrast, parsimonious VIs-based linear regression models are much easier to build, and only small number of field samples and several remote sensing spectral data are required. Consequently, VI-based models have been widely used to estimate the AGB of coastal wetlands in different regions around the world. However, these models have always had low accuracy, with $R^2$ values around 0.40 [16,18–21], and were site-specific. For example, Riegel et al. [16] estimated the AGB of a restored coastal plain wetland using an NDVI-based linear model, which had a $R^2$ value of 0.34. Ghosh et al. [20] developed several MODIS VIs-based (with spatial resolution of 250 m and 500 m) models to estimate the AGB in the northern Gulf of Mexico, which had high $R^2$ values. However, the models could only be applied to tidal wetlands that need to be completely covered by at least 8–10 pixels worth of MODIS images. Doughty and Cavanaugh [21] predicted the AGB of a salt marsh in Santa Barbara County, CA, USA, using a linear regression model based on unmanned aerial vehicles (UAV) image-derived NDVI. Despite the high spatial resolution of the UAV images (6.1 cm), the model did not show good performance (with $R^2$ and RMSE of 0.36 and 496 g m$^{-2}$, respectively). Kulawardhana et al. [19] examined the regression relationships between high spatial resolution digital aerial images based various VIs and AGB in a small salt marsh (~10 km$^2$) and found that the $R^2$ values ranged from 0.14 to 0.28. Miller et al. [18] estimated AGB of two salt marshes using a linear regression model that included both the modified soil vegetation index 2 (MSAVI2) and the visible difference vegetation index (VDVI) from 3 m resolution multispectral data; the model showed good performance in a small salt marsh with area of 29 km$^2$, but this model might have poor performance when applied at a large spatial scale.

Coastal wetlands are often distributed in small patches or narrow strips along coastlines at spatial scales that are often much finer than the resolution of widely used land cover data (e.g., the MODIS based land cover products). Thus, high spatial resolution remote sensing data are required to identify spatial details within coastal wetlands, and robust VIs-based models are needed to accurately and rapidly estimate the AGB of coastal wetlands at large spatial scale and at a low cost. Emergent cloud-computing technologies and services such as the Google Earth Engine (GEE) [22] as well as freely available and high-resolution remote sensing data (e.g., Sentinel satellite data) make these needs possible. The objectives of this study were to develop a reliable VIs-based coastal wetland AGB model by combining Sentinel-1 SAR data, Sentinel-2 images, climate data, topography data, and field AGB samples and to estimate the AGB of the coastal wetlands over the Bohai Rim using the model and the recently developed high-resolution Bohai Rim coastal wetland map (BRCW10) [23].

## 2. Materials and Methods

### 2.1. Study Area

The Bohai Rim (E 116.7°~123.5°, N 35.2°~41.5°), which is located in the coastal zone of northern China (Figure 1), is the political, economic, and cultural center of northern China. In this study, the coastal region of Bohai Rim was defined as the region including the Tianjin municipality and the coastal counties around Bohai. It covers an area of approximately 100, 248 km². Based on temperature and precipitation data derived from the observation-based China climate data set (http://data.cma.cn/en, accessed on 26 September 2021), the mean annual temperature (MAT) and mean annual precipitation (MAP) of the Bohai Rim during 1980–2019 were 11.2 °C and 614.2 mm yr$^{-1}$, respectively. Coastal wetlands are concentrated and provide important ecosystem services in this region. The coastal wetland area is estimated to be approximately 8095.8 km², accounting for about 8.1% of the region [23]. There are diverse types of coastal wetland in the area, including estuary wetlands, tidal flats, salt marshes, riverine wetlands, and constructed coastal wetlands (i.e., paddy, salt field, reservoir, urban wetland, mariculture ponds) (Figure 2). Nevertheless, with the increasing population and urban land use pressures and accelerated climate change that have occurred over the past several decades, coastal wetlands have become seriously degraded. It was reported that the shoreline of the Bohai Sea changed at an average rate of 188.47 m yr$^{-1}$ over the past 30 years due to the intensive land reclamation [24].

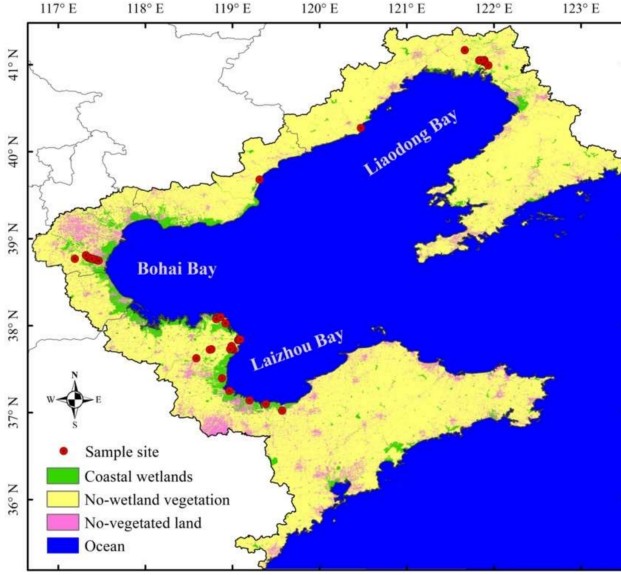

**Figure 1.** Location of the Bohai Rim coastal zone. Spatial distribution of coastal wetlands and AGB field samples used in the study are also shown. The coastal wetland map is based on the BRCW10 data [23].

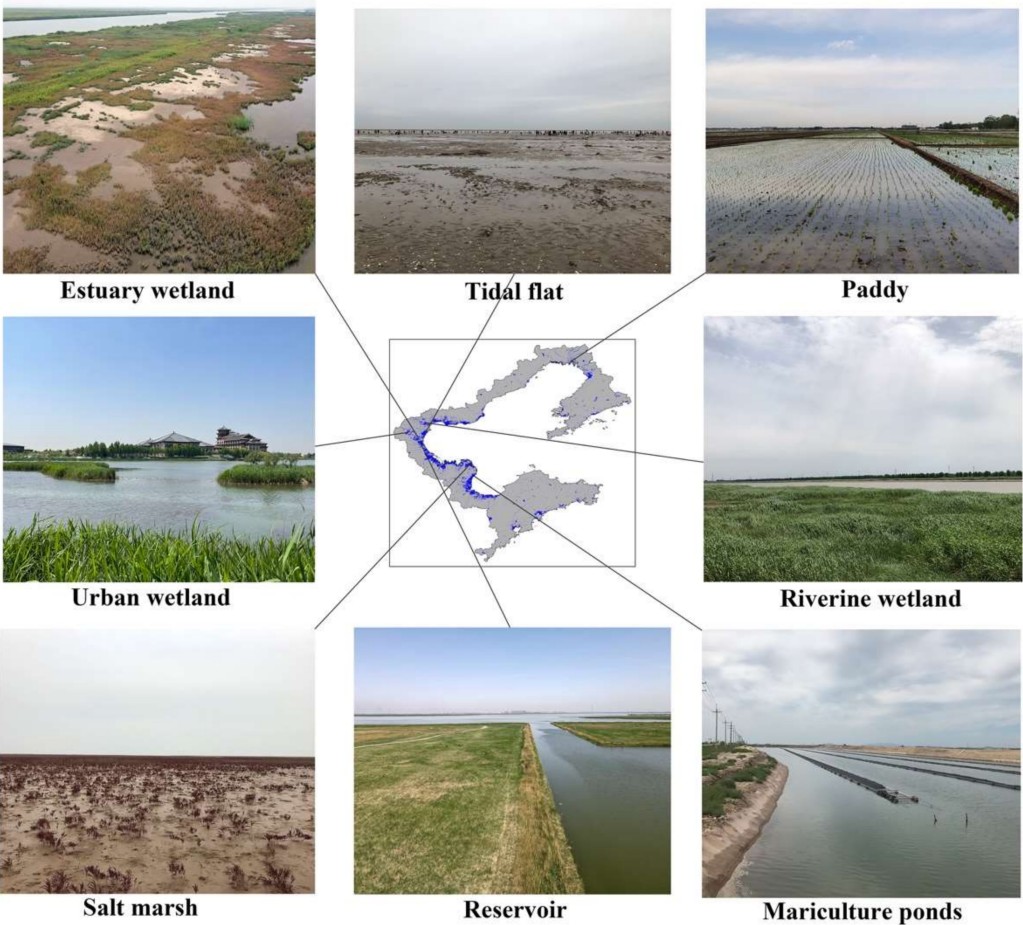

**Figure 2.** Landscapes of the diverse coastal wetlands in the Bohai Rim, China.

## 2.2. Field Sampling

We collected 71 AGB field samples from the Bohai Rim coastal wetlands over the period of 2015–2019 (Figure 1) by compiling 36 AGB samples from published literature and by sampling 35 AGB plots in August–September 2019 (Table 1). The location of each sample plot was recorded with a high precision global positioning system. Within 1 m$^2$ of each sampling site, the vegetation species were recorded, and the aboveground live green biomass of all of the plants was sampled. The fresh plants that were sampled were placed in plastic bags and taken back to the lab. The AGB of each sample was measured after the samples were cleaned and dried.

## 2.3. Coastal Wetland Map and Water Body Data

The coastal wetland map used in this study was based on BRCW10 data (Figure 3a), which were generated from the GEE platform using a machine learning algorithm by integrating open-access synthetic aperture radar (SAR) and optical images from the Sentinel satellites and two terrain indices [23]. It represents the spatial distribution of the coastal wetlands in Bohai Rim in 2019 and has a spatial resolution of 10 m.

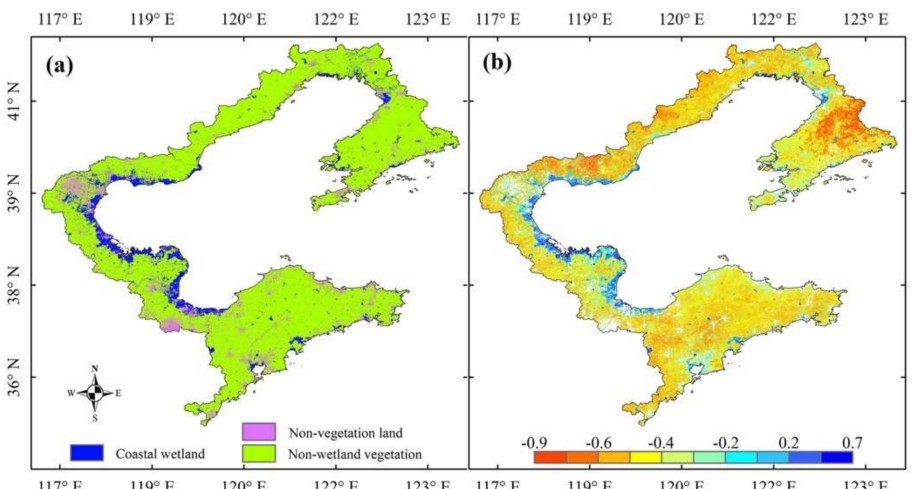

**Figure 3.** The BRCW10 coastal wetland map (**a**) and the normalized difference water index based on Sentinel-2 data (**b**).

To mask out the water areas when modelling the AGB of the coastal wetlands, the normalized difference water index- (NDWI) [25] based surface water body data were used. The NDWI data were calculated as NDWI = (Band8 – Band3)/(Band8 + Band3), using the Sentinel-2 images taken during the growing season in 2019 (Figure 3b). The water body map was derived from the NDWI data with a threshold of NDWI > 3.0 [26].

**Table 1.** Information of the AGB samples.

| Sampling Time | Wetland Type | Vegetation Species | Source |
|---|---|---|---|
| Aug. 2015 [1] | Salt marsh, tidal flats, constructed coastal wetlands | *Suaeda heteroptera Kitag, Phragmites australis* | Yang et al. [27] |
| Aug. 2016 | Salt marsh, tidal flats, constructed coastal wetlands | *Suaeda heteroptera Kitag, Phragmites australis* | Yang et al. [27] |
| May 2018 | Estuary wetland, salt marshes, tidal flats | *Phragmites australis, Suaeda heteroptera Kitag, Tamarix chinensis* | Li et al. [28] |
| Jul. 2018 | Estuary wetland, tidal flats | *Phragmites australis, Suaeda salsa, Tamarix chinensis, Imperata cylindrica, Tripolium vulgare* | Zhao et al. [29] |
| Jul. 2019 | Estuary wetland, tidal flats | *Phragmites australis, Suaeda salsa, Tamarix chinensis, Imperata cylindrica, Tripolium vulgare* | Zhao et al. [29] |
| Aug. 2019 | Estuary wetland, riverine wetland | *Phragmites australis, Suaeda salsa* | This study |

[1] The AGB samples in 2015 were used to approximately represent AGB in the corresponding sites in 2016 because the remote sensing data covering the study area were available from 2016 onward.

### 2.4. Inputs of the Coastal Wetland AGB Model

According to previous studies and the available high spatial resolution remote sensing data, we first determined 23 alternative input variables to build a receivable VIs-based coastal wetland AGB model. The inputs consisted of multiple remote sensing, climatic, and topographic data (Table 2). The remote sensing variables included both Sentinel-1 SAR and Sentinel-2 optical remote sensing data. The topographic characteristics of the region were represented by elevation, the topographic wetness index (TWI) [30], and the topographic position index (TPI) [31]. The climate data included both MAP and MAT.

**Table 2.** List of the independent variables tested in this study.

| Data Source | Independent Variables | Definition |
|---|---|---|
| Sentinel-1 | VV | Vertical transmit-vertical channel |
| | VH | Vertical transmit-horizontal channel |
| | POL | $(VH - VV)/(VH + VV)$ |
| | VVsd | Standard deviation of VV |
| Sentinel-2 | Band 2 | Blue, ~493 nm, 10 m |
| | Band 3 | Green, 560 nm, 10 m |
| | Band 4 | Red, ~665 nm, 10 m |
| | Band 5 | Red edge, ~704 nm, 20 m |
| | Band 6 | Red edge, ~740 nm, 20 m |
| | Band 7 | Red edge, ~783 nm, 20 m |
| | Band 8 | Near infrared, ~833 nm, 10 m |
| | BNDVI | Blue NDVI, $(Band\ 9 - Band\ 1)/(Band\ 9 + Band\ 1)$ |
| | NDVI | $(Band\ 8 - Band\ 4)/(Band\ 8 + Band\ 4)$ |
| | NDWI | $(Band\ 3 - Band\ 8)/(Band\ 3 + Band\ 8)$ |
| | LCI | $(Band\ 8 - Band\ 5)/(Band\ 8 + Band\ 5)$ |
| | EVI | $2.5 \times (Band\ 8 - Band\ 4)/(Band\ 8 + 6 \times Band\ 4 - 7.5 \times Band\ 2 + 10,000)$ |
| | SAVI | $1.5 \times (Band\ 8 - Band4)/(Band\ 8 + Band\ 4 + 0.5)$ |
| | NDSI | $(Band\ 11 - Band\ 12)/(Band\ 11 + Band\ 12)$ |
| Topography | DEM | Digital elevation model |
| | TWI | Topographic wetness index |
| | TPI | Topographic position index |
| Climate | MAP | Mean annual precipitation |
| | MAT | Mean annual temperature |

The Sentinel-1 SAR data used in this study were derived from Sentinel-1 SAR C-band Level-1 Ground Range Detected images [32] and included vertical transmit-vertical channel backscattering (VV) and its standard deviation (VVsd), vertical transmit-horizontal channel backscattering (VH), and the normalized polarization index (POL). The POL was calculated as $(VH - VV)/(VH + VV)$. All of the Sentinel-1 data were accessed and pre-processed with the GEE platform. The data from the ascending and descending orbits both with the interferometric wide swath model and average incidence angles between 30 and 40° were collected.

The VIs and several spectral bands used in this study were derived from the Sentinel-2 Level-1C images [33]. The spectral data included the bands from Band2 to Band8 (B8). The VIs included NDVI, blue NDVI (BNDVI) [34], NDWI, the leaf chlorophyll index (LCI) [35], EVI, the normalized difference salinity index (NDSI) [36], and soil-adjusted vegetation index (SAVI) [37] (Table 2). All of the Sentinel-2 images were acquired and were pre-processed on the GEE platform. The pre-processes included cloud-masking, quality controls, and mosaic. The spectral bands and VIs with a spatial resolution larger than 10 m were resampled to a coincident 10 m spatial resolution. To extract the values of the remote sensing data corresponding to the AGB samples, both the Sentinel-1 and Sentinel-2 images taken during the sampling date and covering the sites were collected. For the entire Bohai Rim region, both Sentinel-1 and Sentinel-2 data during the 2019 growing season were acquired, and a median composite process was applied to each of the remote sensing input variable.

The elevation data were derived from the Global Digital Elevation Model version 2 data (GDEM v2), which is a global open-access digital elevation model (DEM) data with a 30 m spatial resolution [38]. They were generated from the earth's Advanced Spaceborne Thermal Emission and Reflection Radiometer (ASTER) stereoscopic measurements from 2000–2010. The GDEM v2 data were resampled to a 10 m spatial resolution and were used to calculate TWI and TPI. TWI was calculated from the catchment area and slope angel of a special catchment, representing the potential soil water storage condition of a pixel. TPI indicates the elevational position of a pixel relative to the mean elevation of its neighboring pixels. We used the System for Automated Geoscientific Analyses (SAGA) software [39] to calculate TWI and TPI from the re-gridded 10 m spatial resolution GDEM v2 data.

The MAT and MAP data during 1980-2019 were calculated based on the observation-based China climate data set, which was generated by interpolating the observations from 2472 Chinese meteorological stations from 1961 onwards and has spatial and temporal resolutions of $0.5° \times 0.5°$ and month, respectively (http://data.cma.cn/en, accessed on 26 September 2021). To match with the remote sensing and topographic inputs, the MAT and MAP data were resampled to a spatial resolution of 10 m.

We extracted the values of the 23 inputs corresponding to the 71 sampling sites using the extract values to points tool in the ArcGIS software (version 10.5, ESRI, West Redlands, CA, USA, 2016).

*2.5. Evaluation Data*

To evaluate our AGB estimates based on a VIs-based model and multiple data sets, they were compared with the European Space Agency (ESA) Climate Change Initiative Programme (CCI) AGB data [40]. The ESA CCI AGB data were developed to generate the best possible validated AGB maps for climate modelling with existing data. The current CCI AGB version was produced by integrating multiple SAR remote sensing observations, including Envisat ASAR and Sentinel-1 C-band and ALOS-1 PALSAR-1 and ALOS-2 PALSAR-2 L-band data. It has a spatial resolution of 100 m and was acquired for the years 2010, 2017, and 2018. The CCI AGB model created in 2018 was used in this study.

*2.6. Methods*

The multicollinearity of the independent variables often hurts model performance when constructing a model with well correlated variables. To reduce the effects of multicollinearity in the independent variables, Pearson's correlation coefficients (*r*) between AGB and the 23 independent variables and among the 23 variables (i.e., Pearson's correlation matrix) were calculated. For two variables with $r > 0.8$, the *r* values between them and AGB were compared and the one that was less correlated to AGB was removed. After these Pearson's correlation analyses, the variance inflation factors (VIFs) of the remaining variables were examined to further test the collinearity among them using the gvlma package [41] in R software (version 3.5.2, R Core Team, Vienna, Austria, 2018). The variables with a VIF value less than 4 were chosen to fit the VIs-based coastal wetland AGB models [42].

We expected to construct an optimal possible multiple linear regression (MLR) model using the statistically selected independent variables. First, the statistical hypotheses of the independent variables were tested using the gvlma package [41] in R software (version 3.5.2, R Core Team, Vienna, Austria, 2018). Secondly, the QQ plot function in the gvlma package was used to pick out the outliers in the sample data. Finally, the best fitted model was selected using the Akaike information criterion (AIC) method [43], which determined the best fit model by penalizing models with redundant variables.

We validated our best fitted VIs-based MLR model at site scale by against modelling AGB and field measurements. The Person's correlation and root mean square error (RMSE) were calculated. We also compared the Bohai Rim coastal wetland AGB estimates in 2019 based on the model for that year with the CCI AGB model from 2018.

To compare the importance of each independent variable in the VIs-based model, we calculated the relative importance of each variable in the model with a weight function based on the regression coefficients of the model.

## 3. Results

*3.1. Inputs of VIs-Based Model*

According to the Pearson's correlation coefficient analyses (Table S1), 12 input variables were selected from the 23 alternative variables (Table 2). We conducted a few combinations of the 12 selected variables fitting tests and VIF analyses. The results showed that 9 variables including EVI, NDSI, POL, BNDVI, VV, B8, DEM, TPI, and MAP had small collinearity effects for fitting a statistical model and had VIF values that were less than 4. The Sentinel-2 based variables, including EVI, NDSI, BNDVI, and B8 showed significant

differences between coastal zone and inland regions (Figure 4). The EVI values in the coastal wetlands were much lower than those of the other vegetated lands, with values generally being lower than 0.1. The NDSI values ranged from −1 to 1, indicating a large spatial variability in the soil salinity across the region. The BNDVI was <0 in most of the Bohai Rim coastal zone and was significantly lower in the coastal area than it was in the inland region. The spectral values of the B8 were within 153–23,037 and showed significant differences between coastal wetlands and inlands. The VV backscatter data derived from the Sentinel-1 images ranged from −29.6 dB to 29.7 dB and showed significant differences between wetland and non-wetland regions. However, the POL data had small spatial variability. Both DEM and TPI data showed that the topography over the Bohai Rim coastal region is flat, with the exception of several small uplands in the Liaodong Peninsula, western Liaodong Bay, and the Shandong Peninsula. Elevation over the region ranges from 124 m below sea level to 1118 m above sea level. The TPI values were within −83.1–102.2. The MAP ranged from 332.4 mm to 800.7 mm, and there was a significantly variable spatial pattern. This was higher in the Liaodong Peninsula, Liaodong Bay, and southern Laizhou Bay, which had values that were 550 mm higher than those in Bohai Bay and in the Shandong Peninsula.

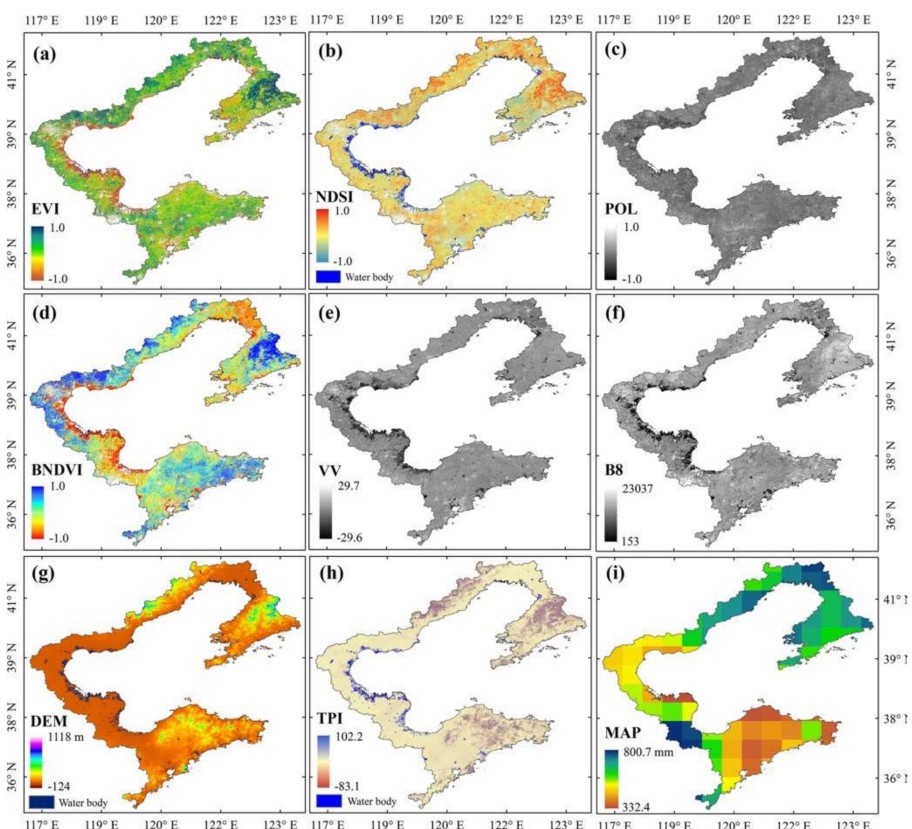

**Figure 4.** Spatial distribution of variables as model inputs after multicollinearity test. (**a**) EVI, (**b**) NDSI, (**c**) POL, (**d**) BNDVI, (**e**) VV, (**f**) B8, (**g**) DEM, (**h**) TPI, and (**i**) MAP.

### 3.2. VIs-Based Coastal Wetland AGB Model

Our statistical hypotheses tests showed that all of the global stat, skewness, kurtosis, link function, and heteroscedasticity assumptions were acceptable. The QQ plot function analyses showed that four samples might be outliers. The four outlier samples were removed, and the relationships between AGB and the nine selected independent variables were examined before fitting the VIs-based AGB models. The EVI and POL were correlated to AGB with $p < 0.1$ (Figure 5). There were significant linear relationships ($p < 0.01$) between AGB and BNDVI and DEM (Elevation). The MAP was significantly correlated to AGB,

with $p < 0.05$. However, there were no significant relationships between AGB and NDSI, VV, B8, and TPI ($p > 0.1$).

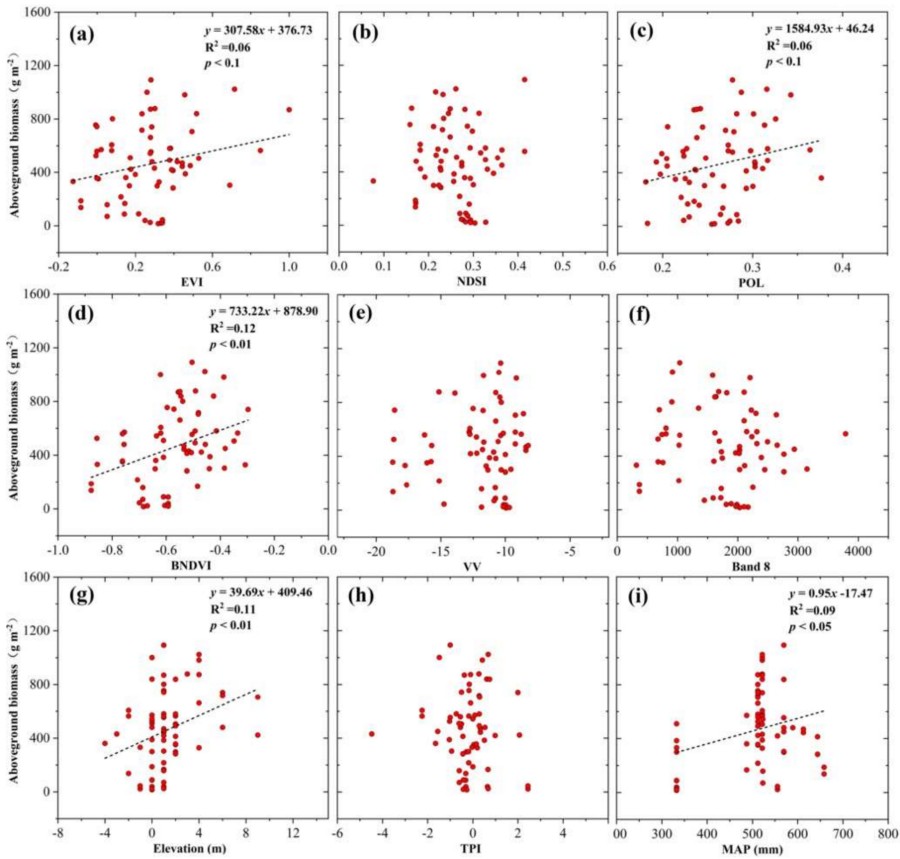

**Figure 5.** Relationships between AGB measurements and EVI (**a**), NDSI (**b**), POL (**c**), BNDVI (**d**), VV (**e**), B8 (**f**), Elevation (**g**), TPI (**h**), and MAP (**i**).

The AIC analyses showed that the best AGB model was a combination of EVI, POL, B8, BNDVI, DEM, TPI, and MAP, and as such,

$$\begin{aligned} \text{AGB} = &\, 426.19 \times \text{EVI} + 1263.5 \times \text{POL} - 0.3 \times \text{B8} + 937.13 \times \text{BNDVI} + \\ &\, 45.22 \times \text{DEM} - 51.65 \times \text{TPI} + 0.61 \times \text{MAP} + 702.24, \end{aligned} \tag{1}$$

The AIC-determined best model was validated using the modelled AGB with the actual AGB (Figure 6). The slope, r, and RMSE values of the linear regression formula were 0.999, 0.74, and 188.32 g m$^{-2}$, respectively.

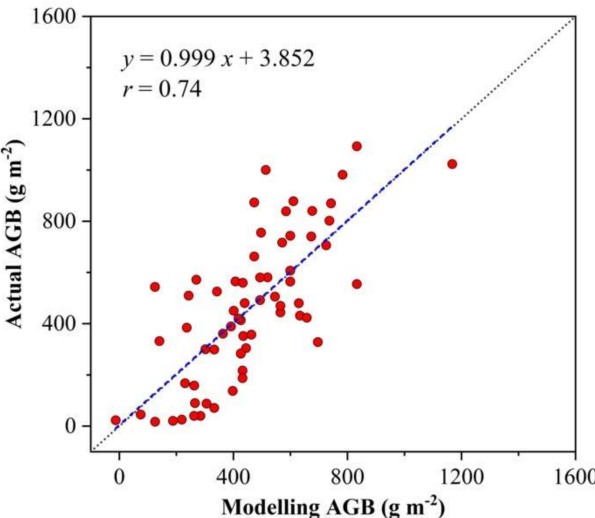

**Figure 6.** Comparison between modelling AGB using Equation (1) and actual AGB.

### 3.3. AGB of the Bohai Rim Coastal Wetlands

Using Equation (1) we estimated the AGB of the coastal wetlands in the Bohai Rim in 2019 (Figure 4). The water bodies in the modelled AGB map were excluded by using a Sentinel-2 derived NDWI based water body map. The final AGB estimate is shown in Figure 7. The final AGB estimation shows that the AGB of the coastal wetlands in the Bohai Rim was within 0–3000 g m$^{-2}$. The AGB map reasonably reflected the spatial variability of the vegetation in the region, which was consistent with our field surveys. For example, in the Beidagang reservoir of the Haihe estuary, the AGB was significantly higher in the western region than it was in the eastern region, which was dominated by water bodies (Figure 7a). The AGB of the coastal wetlands in the Liaohe estuary was higher than that in the Yellow river estuary because of the large nature reserves in the Liaohe estuary and relatively strong human activities in the Yellow River estuary [44,45]. Based on the AGB map, we estimated a spatially averaged AGB of 2.11 Tg C in 2019 in the Bohai Rim coastal wetlands.

We compared our predicted AGB with the CCI AGB during 2018 in the Bohai Rim coastal zone. Consistent with our modelled AGB, the water bodies in the CCI AGB data were excluded with the Sentinel-2 derived NDWI-based surface water body data. It generated a spatially averaged AGB of 9.46 × 10$^{-2}$ Tg C, which was much lower than that from our AGB estimate. The spatial distributions of our predicted AGB and the CCI AGB showed that the CCI AGB might have systematically underestimated the AGB of the coastal wetlands (Figure 8).

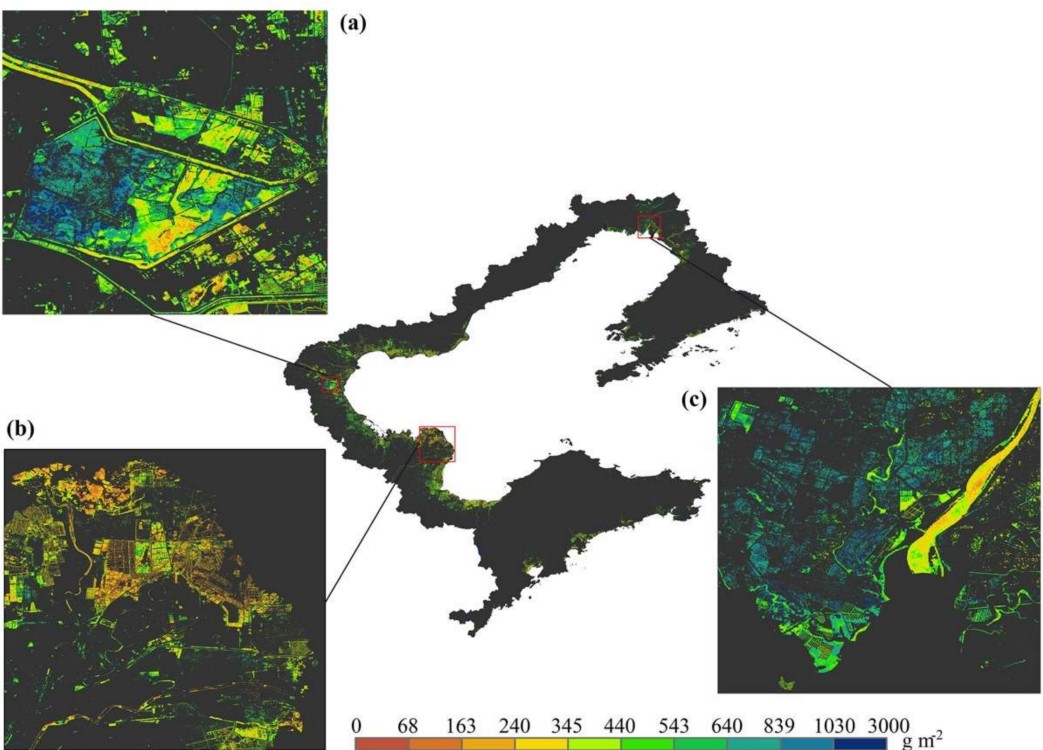

**Figure 7.** Spatial distribution of the coastal wetland AGB in the Bohai Rim in 2019. Spatial details of AGB in the three estuaries of the Bohai Rim are also shown: (**a**) Haihe estuary, (**b**) Yellow River estuary, and (**c**) Liaohe estuary.

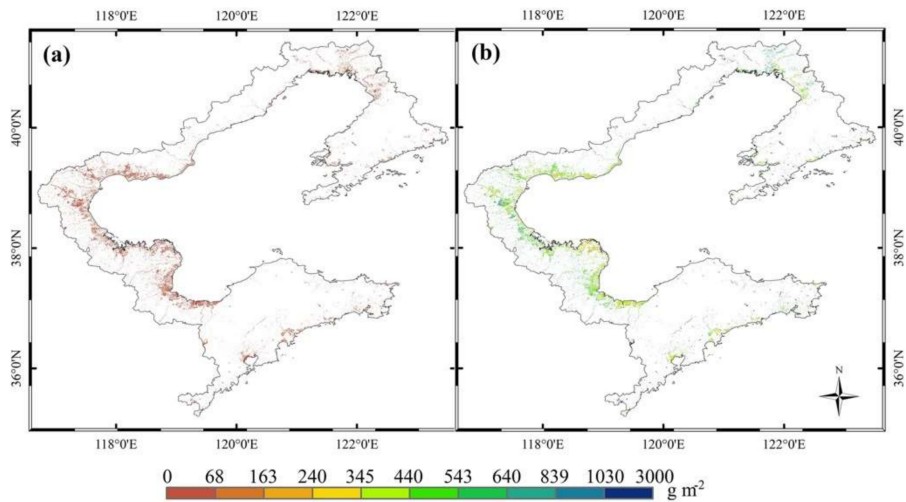

**Figure 8.** Spatial distributions of the AGB of the coastal wetlands of the Bohai Rim derived from the ESA Climate Change Initiative Programme AGB data (**a**) and this study (**b**).

## 4. Discussion

### 4.1. Estimating Coastal Wetland AGB Using Remote Sensing Data

In this study, we estimated the AGB of the Bohai Rim coastal wetlands at a 10 m spatial resolution using a VIs-based model and a recently developed high spatial resolution coastal wetland map. The model was developed by selecting the optimal combination of independent variables from 23 alternative variables, including remote sensing, climate, and topography data. In view of the data, both the VIs-based and ML-based coastal wetland AGB models in most previous studies were based on Landsat or MODIS spectral bands, and they derived Vis with a spatial resolution of 30 m or 500 m. To estimate the AGB of coastal wetlands, remote sensing data with a high spatial resolution were urgently needed because

coastal wetlands are much more fragmented and have much larger spatial heterogeneity when compared to forests, grasslands, and other land ecosystems [18]. As of recently, the open-access and 10 m spatial resolution Sentinel images have made it possible to model AGB at a large scale and at a higher spatial resolution. In addition, the coastal wetland map used to identify the boundaries of coastal wetlands also largely affects estimating coastal wetland AGB with both VIs-based and ML-based models [21]. For a large region, the coastal wetland map used to estimate AGB was often derived from the wetland class of land cover products. The uncertainties in these land cover data-based wetland maps [10] might be an important reason for the low accuracy of the AGB estimate. Thus, it is critical to estimate the coastal wetland AGB to accurately develop a high spatial resolution coastal wetland map [23].

Although ML models were widely used to estimate the AGB of forest and grassland, they were rarely applied to the model AGB of coastal wetlands, possibly because of the difficulties in acquiring sufficient filed samples to be used to train the ML models and the high spatial heterogeneity of coastal wetlands. Two exceptions are the studies of Byrd et al. and Mo et al. [17]. Byrd et al. estimated the AGB of tidal marshes of the conterminous United States based on the random forest algorithm. The random forest model showed slightly higher $R^2$ values (0.58) than our VIs-based MLR model ($R^2$ = 0.55) but had a larger RMSE value (310 g m$^{-2}$). Mo et al. [17] assessed the AGB of coastal marshes in the east Barataria Bay, LA, using both VIs-based and ML-based models, and found that the random forest models performed much better than the VIs-based models (with $R^2$ of 0.84 and <0.1, respectively). The relatively high performance of the ML-based model in these studies could be attributed to both of the overall similarities in vegetation type (i.e., tidal marsh plant function type) across the regions and the advances in the random forest algorithm. In addition to the spatial resolution of the remote sensing data and uncertainties in the coastal wetland map that was used, the relatively low $R^2$ values of the VIs-based coastal wetland AGB models that only rely on several VIs could be also attributed to large differences between field sample scales and remote sensing image, differences in the data acquisition time, and a lack of auxiliary data such as topography, microwave scatter, and climate data in the regression model.

*4.2. Importance of the Independent Variables in the Coastal Wetland AGB Model*

To compare the importance of the independent variables in our VIs-based coastal wetland AGB model, the relative importance of each independent variable was calculated in R software using a weight function according to their regression coefficients. The results suggested that B8, BNDVI, and DEM had much greater predictive power than the other independent variables (Figure 9), with relative importance values of 22.58%, 21.29%, and 19.10%, respectively. For the other variables, EVI had higher relative importance followed by MAP, POL, and TPI, which had values of 12.11%, 11.44%, 8.39%, and 5.04%, respectively. Band8 of the Sentinel-2 images is the near infrared band (NIR), and its spectral reflectance is more sensitive to vegetation type than it is to visible bands [46]. The NIR band has often been used to estimate coastal wetland AGB in both linear regression models [46] and ML-based models [17]. We found that BNDVI was more correlated to coastal wetland AGB than the generally used NDVI and EVI, which is possible because it is more sensitive to the vegetation chlorophyll content and is more helpful for identifying hotspots with low plant photosynthesis [47]. Elevation affects inundation, salinity, and nutrient conditions in coastal wetlands, which determine vegetation distribution and its productivity [8]. For example, *Phragmites australis* has higher AGB and grows in lower water level regions (i.e., higher elevation), while *Suaeda salsa* (with relatively low AGB) widely grows in tidal flats. This study confirmed that elevation factors (i.e., DEM and TPI) are necessary predictors of AGB in coastal wetlands. Consistent with previous studies [10], both C-band VV and VH backscatters from the Sentinel-1 images were not helpful for improving the VIs-based linear regression model, which might be due to their insensitivity to vegetation structure [48]. However, POL benefited the performance of the VIs-based model performance because

it is more sensitive to vegetation structure and dry biomass [49] and is more useful for discriminating between bare and vegetation surfaces [50]. The POL data can also reduce potential outliers within the data by normalizing the depolarization ratio [51] and has valuable potential for improving land-cover classification due to its responsive relationship with plant water content and soil moisture [52]. Previous studies specified that temperature has a significant influence on coastal wetland vegetation [53]. With decreasing MAT, there is often a linear decrease in vegetation productivity [54]. In this study, our analyses suggested that MAT has limited effects on the AGB of the Bohai Rim coastal wetlands, which is possibly because of the small spatial variation in the MAT. This was supported by spatial variation of MAT at the AGB samples, which suggested a standard deviation value of 1.43 °C. In contrast, we found that AGB was significantly correlated to MAP (Figure 5a). This result was consistent with previous review analyses [54] and field experiments [9]. Feher et al. [54] found that in coastal zones, small changes in precipitation can largely affect vegetation growth. Chu et al. [9] conducted a precipitation manipulation experiment in the Yellow river estuary wetland and demonstrated that precipitation promoted AGB by increasing soil water content and decreasing soil salinity.

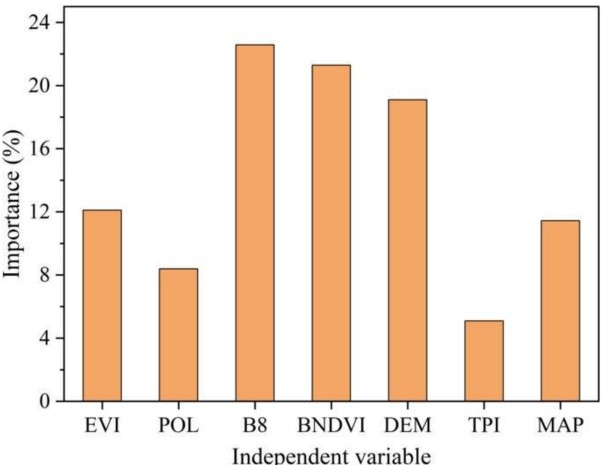

**Figure 9.** Relative importance of the independent variables in the coastal wetland AGB model presented in this study.

Overall, we demonstrated that integrating multi-remote sensing data with a high spatial resolution and several auxiliary metrics can effectively improve VIs-based coastal wetland AGB models. Similar findings have been reported for forests [12], salt marshes [19], and grasslands [11]. These studies have consistently shown that the fusion of multi-source metrics can fit models with higher accuracy than those with VIs alone. In addition, the relatively small size of the AGB samples and the wetland map without wetland subclasses that was used might limit the accuracy of our VIs-based ABG model. We predict that the model accuracy will be increased with more field samples and more coastal wetland class maps.

## 5. Conclusions

In this study, a VIs–based coastal wetland AGB model was developed to estimate the AGB of coastal wetlands in the Bohai Rim. The model showed good performance, with RMSE and *r* values of 188.32 g m$^{-2}$ and 0.74, respectively. The results demonstrated that in addition to high spatial resolution VIs, simultaneously including SAR, topography, and climate data in the VIs-based MLR model can effectively improve the predictive power of the model. The model was based on emerging, open-access, high spatial resolution, multi-remote sensing data, freely available DEM, and climate data. It can be easily applied at a large spatial scale and at a low cost. We used the model to estimate the AGB of the Bohai Rim coastal wetland. The results suggested that AGB in the region ranged from 0

to 3000 g m$^{-2}$; the aboveground biomass C stock over the region was 2.11 Tg C in 2019. The multi-data-based coastal wetland AGB model is valuable for the rapid monitoring of "blue carbon", a special carbon sink, at a large scale. Although the model had a higher R$^2$ value than the VIs-based models seen in most previous studies, it might be still localized and may need to be optimized/modified when being used in other coastal regions due to the various coastal wetland types and the large heterogeneity across regions. Future study should improve the VIs-based model using more field samples and a wetland map with wetland subclasses. The AGB estimate for the Bohai coastal wetlands in 2019 is available at https://figshare.com/articles/dataset/An_aboveground_biomass_estimate_of_coastal_wetlands_over_the_Bohai_rim_China/15140991, accessed on 26 September 2021).

**Supplementary Materials:** The following are available online at https://www.mdpi.com/article/10.3390/rs13214321/s1, Table S1: Correlations among the variables.

**Author Contributions:** Conceptualization, S.S. and Z.S.; methodology, S.S.; software, S.S.; validation, S.S., Y.Z. and Y.W. (Yafei Wang) and W.C.; formal analysis, S.S., Q.L. and L.W.; investigation, S.S.; resources, S.S. and C.C.; data curation, S.S., Q.L. and L.W.; writing—original draft preparation, S.S.; writing—review and editing, Z.S., X.C., W.Y., X.W., X.R. and Y.W. (Yidong Wang); visualization, S.S.; project administration, S.S.; funding acquisition, S.S., Z.S. and Y.W. (Yafei Wang). All authors have read and agreed to the published version of the manuscript.

**Funding:** This research was funded by the Natural Science Foundation of Tianjin City, grant number 20JCQNJC01560; the Peiyang Young Scholars Program of Tianjin University, grant number 2020XRG-0066; and the National Natural Science Foundation of China, grant number 42001131, 41930862, and 41801061.

**Data Availability Statement:** The Sentinel-1 and Sentinel-2 images are freely available from the Google Earth Engine platform. The BRCW10 wetland map is available from https://figshare.com/articles/dataset/A_high_spatial_resolution_map_of_coastal_wetlands_over_the_Bohai_rim/12792626 (accessed on 8 December 2020). The GDEM v2 data are available from https://asterweb.jpl.nasa.gov/gdem.asp (accessed on 31 August 2011).

**Acknowledgments:** The authors thank Shaopan Xia in Nanjing Agricultural University for his help on AGB measurements.

**Conflicts of Interest:** The authors declare no conflict of interest.

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
