# Peer review of "Modelling Aboveground Biomass Carbon Stock of the Bohai Rim Coastal Wetlands by Integrating Remote Sensing, Terrain, and Climate Data"

_remotesensing, doi:10.3390/rs13214321_

Round 1
Reviewer 1 Report
The paper is an enhancement of what already published by the same Group of Research in 2020. The paper is introducing a new parametrization to obtain biomass carbon stock in the Bohai Rim coastal wetland taking advantage of the available free Sentinel-2 data and exploiting the use of heterogeneous data (remote sensing, terrain, climate data). The main drawback of the paper is the section related to Discussion, whose lines 325-364 are more suitable to an introduction and lines 365-376 to a conclusion.
The only Discussion is Section 4.2. Thus, I suggest the Authors to rewrite discussion Section.
In the following some minor remarks:
-line 117: The explanation of the data used to derive MAT and MAP is missing. I suggest to include here lines 205-210 for more clarity.
- line 136: use only 1 m2
-lines 169-170: please check the sentence
lines 231-232: Why do the Authors decide to use this threshold? An explanation is missing.
Reviewer 2 Report
The authors of paper titled “Modelling aboveground biomass carbon stock of the Bohai Rim coastal wetlands by integrating remote sensing, terrain, and climate data” was well written with acceptable results. The authors properly divided the sections and each section was well described. However, this article need some revisions before go for final publication.
The typographical and sentence errors were highlighted in the article.
Some of the major revisions are suggested as follows;
- The abstract section need to be revised more precisely with adopted methods and achieved results.
- Line 30-33: Revise the sentences “Here, we developed ……………………….climate data.” as “In this paper, an efficient coastal wetland AGB model for Bohami Rim coastal wetlands was proposed/presented based on multiple data sets. The model was developed statistically with 7 independent variables from 23 metrics derived from remote sensing, topography and climate data.”
- Line 89: correct “filed” as “field”
- Line 92: Correct “R2”
- Line 91-95: The sentence is not clear. Revise the sentence “However, the models…………………… regions.”
- Section 2.1 Study area: mention the geographic coordinates of the study area i.r. Bohai Rim
- Line 132: correct “filed” as “field”
- Line 241: Correct “Person correlation” as “Pearson’s correlation”
- Line 273: Figure 4: revise the title as “Spatial distribution of variables as model inputs after multicollinearity. (a) ENVI, (b) NDSI………………………………………… (i) MAP.”
- Line 335: correct “filed” as “field”
- Line 339: Correct “R2”
- Line 349-376: This entire section may be shifted to introduction section. It confusing why author again listed many previous related work with references in discussion section. Instead of this. If author discuss about the comparative results of previous studies with present study might be good. – revise this section or move it to introduction section.
- Line 380: revise “R soft” as “R software”
- Line 388: Correct “Different from previous studies, we found that” as “Previous studies states that,”
- Line 395: revise “Our analyses confirmed this knowledge that” as “This study confirmed that,”
- Line 397: delete “we found that”
- Line 400-404: Revise the sentence “It is more sensitive ……………………………………………… soil moisture [51].”
- Line 404: revise “Previous studies suggested ….” as “Previous studies specified ….”
- Line 405-407: The sentence is not clear. “Across…… productivity [53].”
- Line 407: Revise “While, in this study, we found that MAT”
- Line 412-417: The purpose of sentences was not clear here. “Feher et al. ………………………….soil salinity.”
- Line 414: The reference number was missed “Chu et al. [ ]”
- Line 418-420: The sentence is not clear and need to be revised by maintaining continuity of the description. “We demonstrated that ………………………… using VIs.”
- Line 423: delete “Nevertheless, we acknowledge that” and revise the sentence as “The relatively small size of the AGB samples and the wetland map used without wetland subclasses might limits the accuracy of VIs-based ABG model, and predicted that, the model accuracy will be increased with more number of field samples and coastal wetland class map.”
- Line 432: Revise “In this study we developed a VIs-based coastal wetland AGB model to estimate……..” as “In this study, VIs–based coastal wetland AGB model was developed to estimate ……….”
- Over all, the article carried out an efficient work by the authors. But the article need to be revised with proper sentences in all most all the sections. Try to avoid “we” in the article and revise the sentences accordingly. Most of the sentences are complex in nature, and authors can simply of the sentences with same meaning.
- Discussion section need be revised with proper sentences, and efficient discussion of the results.
- Also, revise the conclusion section.
